# MicroRNAs: Their Role in Metastasis, Angiogenesis, and the Potential for Biomarker Utility in Bladder Carcinomas

**DOI:** 10.3390/cancers13040891

**Published:** 2021-02-20

**Authors:** Raneem Y. Hammouz, Damian Kołat, Żaneta Kałuzińska, Elżbieta Płuciennik, Andrzej K. Bednarek

**Affiliations:** Department of Molecular Carcinogenesis, Medical University of Lodz, Zeligowskiego 7/9, 90-752 Lodz, Poland; damian.kolat@stud.umed.lodz.pl (D.K.); zaneta.kaluzinska@stud.umed.lodz.pl (Ż.K.); elzbieta.pluciennik@umed.lodz.pl (E.P.); andrzej.bednarek@umed.lodz.pl (A.K.B.)

**Keywords:** bladder cancer, urothelial bladder carcinoma, metastasis, microRNA, angiogenesis, EMT, adhesion molecules, exosomes

## Abstract

**Simple Summary:**

In cells possessing invasive or metastatic capabilities, several classes of proteins that are involved in the tethering of cells to their surroundings are altered. Loss of cell–cell and cell–matrix adhesions, proteolysis, and induction of angiogenesis are all processes that allow the development of cancer invasion and metastasis. Additionally, epithelial mesenchymal transition (EMT) is a major contributor to the aggressive behavior of cells responsible for invasive growth and metastasis. Here we cover the roles of a few microRNAs (miRNAs) on the molecules that have been shown to play a significant role in bladder cancer metastasis and angiogenesis. These miRNAs could be considered as therapeutic modalities or as diagnostic biomarkers in future research.

**Abstract:**

Angiogenesis is the process of generating new capillaries from pre-existing blood vessels with a vital role in tumor growth and metastasis. MicroRNAs (miRNAs) are noncoding RNAs that exert post-transcriptional control of protein regulation. They participate in the development and progression of several cancers including bladder cancer (BLCA). In cancer tissue, changes in microRNA expression exhibit tissue specificity with high levels of stability and detectability. miRNAs are less vulnerable to degradation, making them novel targets for therapeutic approaches. A suitable means of targeting aberrant activated signal transduction pathways in carcinogenesis of BLCA is possibly through altering the expression of key miRNAs that regulate them, exerting a strong effect on signal transduction. Precaution must be taken, as the complexity of miRNA regulation might result in targeting several downstream tumor suppressors or oncogenes, enhancing the effect further. Since exosomes contain both mRNA and miRNA, they could therefore possibly be more effective in targeting a recipient cell if they deliver a specific miRNA to modify the recipient cell protein production and gene expression. In this review, we discuss the molecules that have been shown to play a significant role in BLCA tumor development. We also discuss the roles of various miRNAs in BLCA angiogenesis and metastasis. Advances in the management of metastatic BLCA have been limited; miRNA mimics and molecules targeted at miRNAs (anti-miRs) as well as exosomes could serve as therapeutic modalities or as diagnostic biomarkers.

## 1. Introduction

Bladder cancer (BLCA) is the most malignant disease of the urinary tract with variable metastatic potential. Categorized as muscle invasive (MIBC) or non-invasive (NMIBC), BLCA is the fifth most common cancer worldwide, resulting in substantial morbidity and mortality of affected individuals [1]. In BLCA, tumor staging presents the depth of bladder wall invasion, with stages Ta and T1 where the tumors are isolated to the urothelium and lamina propria, respectively, considered as NMIBC, while stages T2, T3, and T4, which has invaded the muscle as MIBC. Distinct molecular subtypes have different prognostic, predictive, and therapeutic implications and inform decisions regarding the most appropriate course of action. NMIBC comprises around 70% of BLCA, with five-year recurrence free survival rates of 43% for low risk and 33% for intermediate risk, but up to 21% of high risk progressing to MIBC [2]. NMIBCs is usually nonaggressive, though recurs requiring long term invasive management and surveillance, whereas 75% of newly diagnosed MIBCs do not metastasize; nonetheless, if they do, they become clinically aggressive and invasive with high mortality rate [3]. MIBC is an aggressive epithelial tumor with a high rate of early systemic dissemination, where one third of the patients develop locally invasive or metastatic form [4]. During metastatic dissemination, cancer cells locally invade the surrounding tissue, intravasing the blood and lymphatic microvasculature, surviving and translocating through the bloodstream to microvessels of distant tissues, and adapting to the microenvironment of these tissues. This thus facilitates colonization, proliferation, and the microscopic formation of secondary tumor [5]. Therefore, early detection of BLCA prior to proximal and distal metastasis would immensely improve patient survival and quality of life.

Angiogenesis and inflammation are vital for cancer development, whereby new blood vessels are formed from pre-existing ones. Excessive abnormal angiogenesis is induced due to an imbalance between pro and anti-angiogenic factors, mainly controlled by tissue hypoxia-triggered overproduction of vascular endothelial growth factor (VEGF). Additionally, tumor cells facilitate progression by utilizing different communication strategies for adequate oxygen and nutrient supply [6]. Increased angiogenesis is involved in tumor growth, metastasis, and survival of several tumors including urothelial carcinoma. VEGFA or hypoxia-inducible factor 1α (HIF-1α) are angiogenic factors considered as potential biological markers for BLCA progression to invasive tumors [7]. Exosomes are a subset of extravascular vesicles (EV) that partake in intracellular communication through transferring biologically active molecules including RNA, DNA, and proteins [6]. Tumor cells release exosomes as a means of communicating with neighboring tissues, promoting EMT, angiogenesis, and immune escape [8]. Motz and Coukos state that tumor development requires the development of neovasculature and the suppression of excessive inflammation, thus promoting angiogenesis driving the polarization of the tumor microenvironment (TME). It has been demonstrated that disruption of angiogenesis enhances the efficacy of immune based cancer therapies including vaccines and adoptive cell therapy [9].

MicroRNAs (miRNAs) are a group of highly conserved small non-coding RNAs, with a vital role in regulating the expression of protein coding genes, and could function either as oncogenes (oncomiRs) or tumor suppressors. They can regulate signal transduction pathways in urological tumors, as they have been found to be dysregulated in bladder and renal cell cancers amongst others. miRNAs are key players participating in different stages of the signal transduction process [10]. Their mode of action depends mainly on their target genes possibly exerting tissue or organ specific function. Over 40 miRNAs are involved in urological cancer, and a number of them target common carcinogenic pathways. They regulate cell proliferation, EMT, tumor invasion pathways, angiogenic signaling, and control the expression of more than a third of human genes [11]. Not only do they regulate cell reprogramming, cytoskeletal dynamics, neural development, and plasticity, but they are frequently reported to be downregulated in several types of cancers. They function by targeting several mRNAs affecting a multitude of transcripts to control cellular metabolisms; therefore, their dysregulation influences numerous cancer-relevant processes such as proliferation, differentiation, apoptosis, and metastasis [10]. They function to stabilize mRNA transcripts via post-transcriptional gene silencing through inhibiting the translational process of their target mRNAs via binding partially or fully to complementary sequences of mRNA 3′-untranslated region (3′UTR) [12].

Owing to the genomic events including mutations, deletions amplifications, or transcriptional changes, miRNAs are dysregulated in several diseases including cancer [13]. Several mechanisms can lead to miRNA dysregulation including deletion or amplification of miRNA genes, aberrant expression of transcription factors, and dysregulation of its biogenesis and miRNA sequestration (miRNA sponging) [14]. miRNAs can be found in urine extracellular vesicles and are protected from degradation, making them promising prognostic markers in BLCA [15]. A study identifying miRNAs for NMIBC BLCA diagnosis from blood samples was found to reach an accuracy greater than 90% in the diagnostic index, which could be used to detect the early stage and both low and high grade disease before it metastasizes [16]. Proving miRNAs a noninvasive, effective biomarker for early detection in BLCA.

MicroRNAs exert post transcriptional control of protein expression, leading to protein repression, thus controlling the expression of checkpoint receptors targeting multiple checkpoint molecules while mimicking the therapeutic effect of combined immune checkpoint blockade (ICB) [17]. There is substantial evidence indicating that they and their biogenesis machinery are involved in cancer development. In regards to BLCA impaired miRNA, biogenesis contributes to its clinical progression [10]. Modulating miRNA expression and activity in vivo via miRNA mimics (anti-miRs) presents innovative therapeutic approaches strategies. miRNA mimics have been used to replenish miRNAs with tumor suppressive functions, chemically modified to have higher stability or to enable targeted delivery to tumors [13].

## 2. Pathways Involved in the Metastatic Process and the Role/Importance of Plasticity

The identification of signaling pathways leading to the activation of epithelial to mesenchymal transition (EMT) during tumor invasiveness and other diseases provides insights into the plasticity of cellular phenotypes [18]. EMT and the reverse process mesenchymal to epithelial transition (MET) play a vital role in different steps of metastasis formation, where at distant sites, a more epithelial phenotype is favorable, as cells have to adhere to their surroundings. Therefore, epithelial plasticity is essential for successful metastasis formation, as EMT can provide benign epithelial cancer cells with the required traits to travel to metastatic sites initiating secondary tumor growth. Reactivation of EMT during cancer progression enhances the metastatic phenotype [19]. During EMT, epithelial cells lose their polarity, disassemble cell junctions, and gain migratory characteristics, acquiring a more mesenchymal phenotype. It is important to point out that during EMT there is no change in cell adhesion but rather a fundamental reorganization of cell topology [20]. A number of genes involved in cell adhesion, migration, invasion, and mesenchymal differentiation are transcriptionally altered during the execution of EMT program. Amongst the EMT-inducing transcriptional factors associated with metastasis and tumor invasion are Snail, Slug, dEF1, SIP1, Twist1, FOXC2, and Goosecoid [21]. It is hypothesized, since EMT causes the loss of epithelial characteristics, that it contributes directly to the variation of epithelial cell adhesion molecules (EpCam) and cytokeratin expression seen in circulating tumors. EMT-like phenotypes have been shown to bestow chemoresistance [22]. The functional loss of E-cadherin gene promoter is a hallmark of EMT, which is detected in distant metastases, and in humans the E-cadherin promoter; the E-box elements are responsible for its transcriptional repression in non-E-cadherin-expressing mesenchymal cells. Several signaling molecules are implicated in the EMT process and carcinomas progress, including WNT, TGFβ, and Notch ligands [23]. Other oncogenic pathways include STAT3, PIK3/Akt, ZEB, oncogenic miRNAs, and long non-coding RNAs (lncRNAs) [24]. Carcinoma cells undergoing EMT establish mechanisms for initiating invasive and metastatic behavior for cells of epithelial cancers, generating life threatening manifestations of cancer progression. Additionally, they exhibit heightened resistance to various forms of existing anti-cancer therapies. Hallmarks of the initiation and early growth of primary epithelial cancers encompass uncontrolled epithelial cell proliferation and angiogenesis. During EMT, polarized immotile epithelial cells undergo multiple biochemical changes, enabling them to assume a motile mesenchymal phenotype. Cancer cells expressing the motile mesenchymal phenotype subsequently enter the invasive-metastasis cascade: intravasation, translocating through the bloodstream to microvessels of distant tissue, extravasation, formation of micrometastases, and finally colonization [5], hence enhancing migratory capacity, invasiveness, resistance to apoptosis and production of extracellular matrix (ECM) components.

Amongst the several molecular processes involved in initiating and enabling EMT completion are the changes in the expression of specific miRNAs, which could be used as a biomarker to demonstrate the cell passage during EMT, possibly providing a lead for intervention [18]. Chaffer et al., 2016. suggested that since cancer cells can reside in various phenotypic states along the EMT spectrum, and the fact that they can transition dynamically between the epithelial and mesenchymal phenotypes manifests their ability to survive and seed metastatic deposits, providing cancer cells with tumor initiating potential and the ability to maintain phenotypic plasticity [23]. This thus promotes successful completion of the metastatic cascade outlined in Figure 1. The ability of cancer cells to perform the metastatic cascade corresponds to the transition between the epithelial, hybrid E/M, and mesenchymal states. EMT promotes tumor aggressiveness by providing plasticity. Lately, there is evidence that carcinoma cells residing in a partially epithelial and partially mesenchymal state, i.e., a hybrid favor tumor progression and metastasis. That is because cancer cells at the metastatic cascade can reverse their phenotype by MET and form secondary tumors. A hybrid E/M state with both features increases the possibility of cells to acquire stem-like properties [25]. Such cells have enhanced tumor-initiation potential and can integrate the properties of cell–cell adhesion and motility facilitating collective cell migration, forming clusters of circulating tumor cells (CTCs) accelerating the metastatic cascade even further. Transcription factor NRF2 can stabilize a hybrid E/M phenotype and even prevent a complete EMT. Its knockdown in RT4 BLCA cell line downregulates E-cadherin and ZEB-1, while its overexpression upregulates E-cadherin and ZEB-1, promoting a more hybrid E/M phenotype. Therefore, NRF2 is a hallmark of the hybrid E/M with maximal propensity to generate metastases, but NRF2 signaling could have different effects depending on the position of the cells in question in the EMT spectrum [26].

### 2.1. Molecular Regulation of Tumor Microenvironment

In 2011, Hanahan and Weinberg proposed that along with genome instability and inflammation, evading immune destruction is another emerging cancer hallmark. They also described that tumors exhibit yet another dimension of complexity by containing a repertoire of recruited ostensibly normal cells ascribing the acquisition of hallmarks traits by creating the tumor microenvironment (TME) [27]. The EMT program in the context of carcinoma pathogenesis is activated in epithelial cells from previously latent EMT programs within individual tumor cells, i.e., TME. The most prominent conduit for EMT activation is the heterotypic interactions occurring between carcinoma cells and those residing in the TME [23]. The signals provided by the primary TME are vital modulators of the invasive and metastatic ability of tumor cells [28].

Intravasation platelets are the first blood cells to interact with tumor cells, and it has been reported that activated platelets contribute to tumor cell proliferation and metastasis, as they release angiogenic, mitogenic proteins, and growth factors within the microenvironment to promote tumor cell growth and invasion. Additionally, platelets facilitate tumor cells adhesion to vascular endothelium, by the formation of platelet-tumor cell heteroaggregates by integrin αIIbβ3 (glycoprotein IIb/IIIa), forming a physical shield around tumor cells, protecting circulating ones from immune mediated lysis by natural killer (NK) cells. Reduced number of platelets or those with defective functions in transgenic mice models were found to associate with decreased metastasis formation [29]. Hence, consistent with their role in cancer development, progression, promoting tumor survival, and metastatic process. The pro-metastatic effect of platelets is mediated via activation of TGFβ and NF-KB signaling pathway. Abrogating TGFβ signaling in tumor cells or its expression in platelets is enough to hinder metastasis and EMT, since platelets are an important source of bioavailable TGFβ for tumor cells during circulation and extravasation. Hynes et al. found that activation of TGFβ alone without the presence of platelets was unable to produce effects with a magnitude similar to that of platelets [28].

#### 2.1.1. Cadherins and Catenins

Cadherins are calcium-dependent transmembrane glycoproteins that are prime mediators of cell–cell adhesion found at the adherens junction in epithelial tissues essential for cell polarity and migration [30]. They comprise cytoplasmic domain anchored to the cell cytoskeleton by catenin family members (α, β and γ-catenin, and p120) for full functionality and extracellular and transmembranous domains, with a specific pattern of expression throughout multicellular organisms [20]. E-cadherins enhance epithelial polarity and have a central role in suppressing the invasive phenotype of urethral bladder cancer (UBC) cells [30]. E-cadherin can negatively regulate mitogenic signaling through interaction with EGFR. Additionally, loss of E-cadherin potentiates Wnt signaling, and its expression can be inhibited by a number of zinc family transcription factors as well as miRNAs [20].

“Classical” cadherins (N- or P-cadherin) are transmembrane cadherins that cooperate with catenin family members through their cytoplasmic domain linking the actin cytoskeleton. In the course of EMT, E-cadherin cleavage destabilizes the adherens junction, releasing β-catenin, which functions as a transcriptional activator for cell proliferation. The canonical Wnt pathway is highly conserved however frequently altered in a number of human malignancies with cadherins acting as its negative regulators. It promotes EMT, increases the expression of N-cadherin, and decreases that of E-cadherin. Wnt/β-catenin plays an important role in the development and promotion of EMT and metastasis. Furthermore, EMT is induced when Wnt/β-catenin interacts with T-cell factor (TCF). Since β-catenin/TCF encode c-MYC protein, the sequestering of β-catenin and the resulting activation of Wnt pathway could cause tumorigenesis [31]. In MIBC tumors, as grade and stage progress, E-cadherins expression levels decrease, accompanied with an increase in expression levels of classical cadherins. This phenomenon is described as cadherin switching and possibly changes intracellular signaling; thus, it presents a key step in invasion and represents and vital aspect of EMT. Meanwhile, N-cadherin interacts with FGFR, causing an activation of the receptor with enhanced downstream signaling, leading to increased cell motility and Matrix metalloproteinases (MMP) secretion [20]. A recent study assessing the role of N-cadherins in patients with invasive BLCA who had undergone radical cystectomy found N-cadherin expression to be observed in 43.7% of the patients significantly associating with advanced pathological stage, lymph node metastasis, and worse recurrence free survival [32]. Additionally, N-cadherin and β-catenin link to platelet-derived growth factor receptor a phenomenon/complex contributing to tumor migration [20]. Hence, the classical cadherins promote a more invasive and malignant urethral bladder cancer (UBC) phenotype. It is well established that the expression of E-cadherin is decreased in metastatic cancers. In fact, during EMT loss of homotypic adhesion results from downregulation of surface E-cadherin expression, which is because E-cadherins play a central role through interaction with β-catenin and the actin cytoskeleton (Table 1). Loss of E-cadherin reprograms global gene expression. This promotes delocalization of β-catenin, which is a transcriptional co-activator from the nucleus to the plasma membrane [33].

##### microRNA, Cadherins, and Catenin in BLCA

Recently, low expression levels of miR-373 and E-cadherins were found in BLCA tissues and cell lines, correlating with stage, grade, lymph node metastasis, and poor overall survival (OS). In EJ and 5637 cells, E-cadherin knockout induces stronger proliferation ability and cell motility, playing a vital role in the miR-373 suppressed EMT signaling pathway, while its deletion associates with tumor recurrence, metastasis, and poor survival of BLCA patients. miR-373 activates E-cadherin expression in BLCA cells and interacts with E-cadherin gene promoter. miR-373 inhibits BLCA proliferation through activating E-cadherin expression, miR-373 expression associated positively with E-cadherin expression, and both their reduced expression correlated with poor OS. In EJ and 5637 cells, miR-373 acted as a tumor suppressor during BLCA progression by recovering E-cadherin expression resulting in decreased levels of CyclinD1, c-MYC, and MMP2 mRNA levels. This thus inhibited proliferation and migratory and invasive capacity. The effects of miR-373 on migration and invasion in BLCA cells were tested using wound healing assay, indicating a slower wound closing in comparison to the control group. Alongside, a transwell assay was used to evaluate the capability, displaying a potent inhibition on migration of the cell lines compared to the controls. Depletion of E-cadherin was found to restore cell invasion and migration compared with sole transfection. Hence, by inducing E-cadherin expression in BLCA cells, miR-373 could attenuate migration and invasion ability. In the same study, less metastasis occurred in lung tissue of Lenti-miR-373 nude mice following miR-373 upregulation, and less metastatic nodules in the liver were found compared to Lenti-dsControl group. This indicates that by activating E-cadherin expression, miR-373 inhibited BLCA cell proliferation and metastasis in vivo and in vitro [34].

Increased invasion and migration ability of CRL1749 and HTB9 cells has been associated with loss of expression of miR-141 and miR-200b. miR-141 and miR-200b play vital roles in the invasive ability and EMT phenotype on BLCA; additionally, their levels of expression were also found to predict lymph node metastasis. In CRL1749 cell line, increased expression of both miRNAs reduced the activity of MMP-2 and MMP-9 enzymes, while the opposite effect was observed in HTB9 cells. MMP-16 is a direct downstream functional target of miR-200b, with the ability to degrade some matrix molecules directly and activate MMP2 and MMP-9 [35], whereas the transmembrane N-cadherin is implicated in promoting cancer cell motility, migration, and invasion and acts as a signal transduction receptor, facilitating cell–cell adhesion [43]. In CRL1749, E-cadherin expression levels significantly increased after miRNA-141 and miRNA-200b were overexpressed, whereas the expression of the mesenchymal markers vimentin and N-cadherin was downregulated. Contrary to that, in HTB9, when both miRNAs’ expression was decreased, E-cadherin mRNA and protein expression levels decreased significantly, while that of vimentin and N-cadherin increased [35]. Together this suggests that modulation of these miRNAs leads to different changes in invasive ability, which is associated with the EMT phenotype of these BLCA cell lines, HTB9 cells being a grade II carcinoma and CRL1749 being a transitional cell carcinoma. The regulatory mechanism of miR-145 and N-cadherin in the migration and invasion of BLCA cells revealed that N-cadherin was a direct target of miR-145 and regulating it and MMP9 expression levels, through which it possibly inhibits migration and invasion of BLCA cells. Thus, in BLCA, miR-145 has tumor suppressive effects [36].

As mentioned previously, a number of miRNAs target common carcinogenic pathways. The role of miR-135a was examined in a study recruiting 165 patients where the tumor and adjacent normal tissue of patients who had undergone bladder resection of all tumor grades were paired, both of MIBC and NMIBC. Following EMT induction and targeting GSk3β via the Wnt/β-catenin signaling pathway, miR-135a then accelerated EMT, invasion, and migration of BLCA cells. This was done via activation of Wnt/β-catenin pathway and downregulation of GSK3β, causing increased miR-135a, β-catenin, cyclinD1, vimentin mRNA, and protein expression, whereas decreased GSk3β, E-cadherin mRNA, and protein expression in BLCA tissues compared to the adjacent normal tissue [37]. Ergo β-catenin relates to BLCA progression, and the increase in levels of vimentin accompanied with reduced E-cadherin expression levels following miR-135a overexpression signify its role in EMT regulation, consequently promoting cell proliferation, invasion, and migration.

It is well established that the miR-200 family members have an imperative role in the suppression of EMT, cell adhesion, migration, invasion, and metastasis. miR-200 family were expressed in BLCA cell lines displaying an epithelial phenotype and thus are crucial regulators and modulators of EMT. They are downregulated in human cancers due to the aberrant epigenetic gene silencing. Adam et al. aimed to determine the role of members of miR-200 family in controlling EMT and EMT-induced resistance to EGFR therapy in human BLCA. Higher expression levels of miR-200c seemed to be inverse to those of the EMT genes: ZEB1 and ZEB2, which are direct targets of miR-200 and ERRFI-1. The down-regulation of these three genes is a signature of an EGFR-sensitive BLCA phenotype. Additionally, the expression of miR-200 induces a MET phenotype and reduced ERRFI-1 expression. The expression of miR-200 c and b and modulating its direct or indirect targets including E-cadherin and ERRFI-1 was adequate to reverse the biology of a mesenchymal cell line via EGFR signaling [38]. Mao et al. report a role for GSK-3β in BLCA, as its inactivation from an active EGFR and its downstream PI-3Kinase and MAPK signaling pathways defines the EGFR sensitivity. They also report an association between E-cadherin levels and high anti-EGFR sensitivity, hypothesizing that the spatial distribution of EGFR may play a role in EGFR responsiveness, as EGFR is promptly recycled into the cell membrane of mesenchymal cell lines with E-cadherin serving as a scaffold for EGFR [37]. Accordingly, it might cause increased stability and exposure on the cell surface for optimal accessibility of EGFR-targeted therapy [38]. Another study also showed how metastasis in vitro was inhibited by miR-3619 through interaction with p21 promoter specific sequences, with β-catenin and CDK2 as direct targets of miR-3619. miR-3619 prevented β-catenin accumulation in the cytoplasm or its translocation into the nucleus in BLCA cells, and its overexpression enhanced E-cadherin, the epithelial biomarker, but suppressed expression of mesenchymal biomarkers Vimentin, Snail, and N-cadherin. Moreover, in T24 and 5637, miR-3619 caused marked changes in Cyclin D1 and E-cadherin, p21 downstream genes, downregulating β-catenin and CDK2 expression, consequently inhibiting the EMT process [39].

#### 2.1.2. Integrins in BLCA

Integrins are transmembrane glycoproteins formed from different α and β heterodimers and mediate adhesion to the ECM and immunoglobulin superfamily molecules. They are a large family of ECM receptors, involved in cell–cell and cell–matrix interactions with a prominent role in intracellular signaling, cytoskeletal organization, and cell adhesion and migration [44]. As they directly mediate adhesin to ECM proteins, they regulate several cellular processes and are vital for cell migration and invasion (Table 1). They function as receptors for ECM proteins, specifically laminin and collagen VII, that regulate tissue homeostasis, organ development, inflammation, and disease [45]. Cellular interactions with ECM could result in adhesion mediated drug resistance. They can also modulate several biological behaviors contributing to cancer development or drug resistance [40]. Several studies reported that integrins participate in tumor progression; others have reported their role in the development of resistance to chemotherapy in BLCA with alterations of integrin expression modifying adhesive and invasive behavior of BLCA cells [46].

In tumor cells, integrins function as membrane surface receptors that transduce extracellular signals mediating downstream pro-survival signaling pathways. Integrinα2β1 was reported to facilitate tumor cell migration via EMT facilitating invasion and metastasis in breast cancer. In vivo BLCA expression of integrin-β8 promotes tumor growth and drug resistance development. In BLCA, overexpression of integrin-β8 serves a vital role in cancer cell proliferation and drug resistance development, whereby its blockade notably improved anticancer effects of chemotherapy [47]. Loss of expression of α6β4 integrin could predispose BLCA cells to invasion and metastasis. α6β4 integrin is a component of the hemidesmosomal anchoring complex, and in normal epithelium it colocalizes in close relationship with collagen VII. In human cancers, it has been shown to restrict cell migration via stable anchoring contacts with laminin [48]. Integrins act as activators of several FAK signaling pathways that serve to regulate cell migration and invasion [49]. Integrins can regulate EMT by unsettling cell adhesion and stimulating EMT related gene expression, and those that contain the αV subunit can activate TGF-β inducing EMT. Simultaneously, TGF-β signaling can activate and upregulate integrin expression, with this cross-talk occurring downstream of initial receptor activation overriding the tumor suppressor function of TGF-β [50]. Additionally, there is accumulating evidence that integrins have a vital role in the initial phase of tumor dissemination, which requires EMT forming an important aspect of the metastatic process [41].

##### Integrins and miRNA in BLCA

It has been well established that integrins are known to be regulated by miRNAs, and so the miRNA/integrin axis plays a critical role in cancer biology and the metastatic cascade.

Integrin-β1 (ITGB1) in UBC is responsible for chemotherapeutic drug mitomycin-C (MMC) resistance. In UBC, breast, esophageal, and gastric cancers miR-31 disrupted the migratory and invasive processes. miR-31 expression in UBC was found to correlate with individual prognosis. In non-invasive cases, its downregulation correlated with unfavorable prognosis, and the invasive subtype expressed lower levels of miR-31 compared to the non-invasive phenotype. In vivo and in vitro experiments in UBC indicate that miR-31 heightened sensitivity of UBC to MMC via suppression of ITGA5, an important subunit of fibronectin receptor-integrin α5β1 and downstream pathways. ITGA5 mRNA has one theoretical miR-31 binding site with luciferase reporter assays confirming that ITGA5 is a direct target of miR-31. ITGA5 controls focal adhesion kinase/Akt pathway, a cascade of cell survival, proliferation, migration, and invasion acting as an oncogene. In vitro miR-31 acted as a UBC suppressor through negative regulation of cell cycle, migration, and invasion. Additionally, UBC cells transfected with miR-31 mimics lost viability after 48 h, and its overexpression suppressed proliferation by blocking G1 to S phase cell cycle transition via targeting ITGA5. Additionally, miR-31 overexpression inhibited migration and cell invasion. Moreover, miR-31 expression reversed adhesion mediated drug (MMC) resistance via downregulation of ITGA5 and inactivating Akt and ERK pathways. Overexpression of miR-31 along with MMC treatment resulted in the strongest effect of suppressing tumorigenicity [46]. In BLCA cells, miR-124-3p induced apoptosis and inhibited proliferation, migration, and invasion. Methylation of miR-124-3p contributes to its downregulation and that of several signaling pathways it modulates involved in cell migration and invasion. ITGA3 serves as a direct target of miR-124-3p and inhibits ITGA3 by suppressing FAK phosphorylation, reducing phosphorylated levels of PI3, AKT, and Src. Therefore, miR-124-3p inhibits FAK/PI3K/AKT and FAK/Src pathways. In addition, overexpression of miR-124-3p and knockdown of ITGA3 resulted in upregulation of E-cadherin and downregulation of N-cadherin. Further silencing of ITGA3 enhances apoptosis reducing viability, proliferation, migration, and invasion. Thus, in BLCA cells, miR-124-3p may inhibit migration and invasion through modulating EMT progress via ITGA3 [49].

#### 2.1.3. CD44 in BLCA

CD44 is a ubiquitous cell surface adhesion molecule implicated in cancer development and metastasis, promoting invasion and angiogenesis [48]. It is an ECM receptor involved in cell migration, adhesion, and cellular interactions [44]. Wu et al. previously reported that CD44 is a predictor for radiation response and outcomes of patients treated with concurrent chemoradiotherapy [51]. In invasive BLCA cells, CD44 is divided into standard and variant isoforms, expressed in mesenchymal and epithelial cells, respectively. CD44v6 has the ability to interact with receptor tyrosine kinase, c-Met, in parallel with EpCAM expression, giving cancer cells growth factor microenvironment susceptibility. It is found that increased invasive ability of CD44 is due to increased Akt signaling pathway; furthermore, EMT alters CD44 splicing pattern from CD44v to CD44s [52]. A recent study reported that in MIBC staining of CD44 was significantly associated with clinical lymph node involvement and locoregional recurrence, correlating with lower disease-free survival (DFS) for all MIBC patients. In vitro CD44+ isolated cells had higher invasive ability compared with CD44- ones. However, it is observed that in vivo results in an orthotopic tumor model; CD44 increased invasiveness and metastatic potential of cancer cells compared to the subcutaneous tumor model. In both in vivo and in vitro studies, CD44+ cells exhibited increased invasion related factors and attenuated epithelial characteristics. CD44 cells had a higher capability of invasion and were associated with increased expression of EMT and invasion related factors. Increased levels of IL-6 have been observed with BLCA patients, and an activated cytokine IL-6 signaling provides a suitable microenvironment for CD44 induction where in immunocompetent mouse models blockade of IL-6 decreased CD44 expression attenuating tumor aggressiveness [53].

##### CD44 and miRNA in BLCA

In BLCA, CD44 has been identified as a direct target of miR-34a (Table 1). Expression of miR-34a was found to be downregulated in BLCA cells and bloodstream and plays a vital role in BLCA recurrence and progression. miR-34a interacts directly and specifically with the target site in the 3′UTR of CD44, and miR-34a overexpression increased that of E-cadherin but decreased that of N-cadherin, Vimentin, and β-catenin. Following the observation that overexpression of miR-34a reduced the metastatic nodules in the liver and lungs, it could function as an anti-metastatic microRNA directly targeting CD44. Additionally, in vivo overexpression of miR-34a attenuated metastasis and angiogenesis of BLCA cells promoting E-cadherin expression, however inhibiting N-cadherin, Vimentin, and β-catenin expression, therefore possibly inhibiting EMT and acting as a tumor suppressor. In in vivo and in vitro studies in BLCA xenograft tumors, transfection by miR-34a reduced VEGF and CD44 levels, subsequently reducing angiogenesis. Hence, there miR-34a suppresses angiogenesis via directly targeting CD44 in BLCA, and restoration of CD44 could rescue anti-tumor effects of miR-34a. Circulating levels of miR-34a associate with miR-34a levels in BLCA cells and accordingly could possibly be used as a biomarker [52].

### 2.2. The Interplay between Extracellular Matrix (ECM) and miRNAs

ECM regulates several processes including cell adhesion, proliferation, migration, survival, differentiation, and organogenesis. The synthesis and turnover of ECM affects cell behavior, ergo requiring a balance to maintain normal functioning preventing cellular and tissue changes that can lead to the development or progression of diseases. miRNAs are one of the regulatory mechanisms that control ECM composition, differentiation, and the behavior of residing cells. miRNAs are key regulators of ECM gene expression, contributing to its synthesis, maintenance, and remodeling during development and disease. They can control the composition of ECM and affect the expression of specific miRNAs (Table 2). miRNAs can be regulated by ECM molecules including integrins, CD44, and its ligand hyaluronan (HA). It has been suggested that miRNA biogenesis at both the transcriptional and post-transcriptional level can be affected by ECM via the Drosha- and/or Dicer-mediated interaction. miRNAs can regulate ECM directly via targeting ECM molecule mRNAs or indirectly by modulating the expression of genes regulating the synthesis or degradation of ECM molecules. However, in most cases, direct miRNA targets are transcription factors, e.g., SOX9, growth factors, e.g., BMP7, or signaling molecules, e.g., IKK-β; all implicated in the regulation or synthesis of ECM components. The link between ECM and cancer is recognized where primary cancer cells metastasize through the lymphatic system or bloodstream, thus making ECM, a component of the TME, vital for oncogenesis, cancer progression, and metastasis, since it affects cell growth, survival, adhesion motility, and invasion [44].

#### Matrix Metalloproteinases (MMPs)

Matrix metalloproteinases (MMPs) are calcium dependent zinc containing endopeptidases with the ability to degrade several ECM proteins. They are involved in the cleavage of cell surface receptors and presumably facilitate tumor cell invasion and metastasis through ECM degradation [1]. In developing tissues and organs, MMPs and tissue inhibitors of metalloproteinases (TIMPs), specific endogenous MMP inhibitors, regulate collagen deposition, with miRNAs contributing to the regulation of ECM proteins expression during chondrogenesis [45]. MMP2 and MMP9 promote endothelial cell migration and blood vessel formation [54]. ADAM9 is membrane anchored protease involved in cell–cell and cell–ECM interactions and is a direct target of miR-142-3p, with its expression regulated in chick wing and leg mesenchymal cells, contributing to the modulation of position-dependent chondrogenesis. Experimental evidence seems to suggest that ADAM9 might mediate cell–cell communication for cell survival, as it inhibited cell migration and increased apoptotic cell death. Other ECM molecules including MMPs, aggrecan, and certain types of collagen are indirectly modulated by several miRNAs through SOX9 regulation during chondrogenesis. For instance, miR-145 directly targets SOX9 suppressing chondrogenic differentiation of murine embryonic mesenchymal cells. Overexpression of miR-145 in human normal articular chondrocytes reduced SOX9, COL2A1, and aggrecan levels but increased the hypertrophic markers RUNX-2 and MMP-13. Other examples of miRNAs regulated by SOX9 include miR-675 and miR-1247 during chondrocyte differentiation, affecting COL2A1 production [44].

A hallmark of the EMT process is the loss of E-cadherin expression and an increase in the expression of classical/mesenchymal cadherins (N- or P-cadherins), resulting in enhanced cell motility and hijacking EMT to ensure their migration. In addition, its downregulation changes the morphology and motility of epithelial cells to allow them to adopt mesenchymal characteristics [38]. Loss of E-cadherin along with increased expression of MMPs together served as a better predictor of prognosis rather than the downregulation of the former and the upregulation of the latter separately [34]. In BLCA cell lines HTB9 and T24, TGF-β1 induced EMT by reducing the expression of miR-200, which directly targets MMP-16, in return downregulating its expression. Following TGF-β1 treatment, E-cadherin levels were progressively reduced, while N-cadherin and vimentin expression levels were increased. Additionally, following 12h of TGF-β1 treatment, expression levels of miR-200b were reduced gradually, implying that miR-200b is a metastasis-inhibiting miRNA in BLCA, which was confirmed when miR-200b overexpression inhibited TGF-β1 induced MMP-16 upregulation and cancer cell migration [42]. In UBC 5637 cell lines, miR-370-3p directly repressed Wnt7a expression, suppressing invasion. Thus, miR-370-3p/Wnt7a axis controls UBC invasion through canonical Wnt/β catenin signaling, where Wnt7a activates canonical Wnt pathway to induce EMT, the expression of MMP1 and MMP10 [42]. MMP11 was also identified to be the direct target of miR-139-5p and miR-139-3p. In breast cancer, high MMP11 expression correlated with poor prognosis, and in colorectal cancer, it was a predictive survival marker. Both miRNAs were downregulated in BLCA; their restoration significantly inhibited BLCA cancer cells viability migration and invasion through targeting MMP11. The miR-139-5p/miR-139-3p/MMP11 axis seems to regulate several genes that contribute to cancer cell aggressiveness including CXCL1 and CXCL3 [55].

### 2.3. Mechanism of Metastatic Inhibition or Induction in BLCA Via miRNA

There are many lines of evidence demonstrating the involvement of miRNAs in signaling pathways of proteins regulating metastasis in BLCA (Table 3). Onco-suppressor miRNAs are downregulated in BLCA cells and tissues to ensure proliferation and migration. In the previous sections, we discussed specific miRNAs dysregulation in regards to cell–cell and cell–matrix adhesions, here we will be covering alterations of some direct miRNA target genes and pathways that enhance metastasis in BLCA, as shown in Figure 2.

The expression of most of the below mentioned miRNAs was found to be downregulated in BLCA cells. In cancers, miR-124 is involved in epigenetic regulatory programs with DNA methylation and chromatin remodeling and is also responsible for transcription factor SOX9. In BLCA, UHRF1 is a direct target of miR-124 downregulation. Overexpression of miR-124 in T24 cell line resulted in reduced levels of VEGF, MMP-2, and MMP-9. Additionally, overexpression of miR-124 in vitro attenuated cellular proliferation, migration, invasion, and angiogenesis, while in vivo tumor growth downregulated UHRF1, showing an inverse relationship between miR-124 and UHRF1. In T24 and J82 cell lines, silencing UHRF1 suppresses migration and hinders invasion, thus having an oncogenic role in BLCA, with a role in angiogenesis [56]. By regulating UHRF1, the pre-miR-145 and guide strand miR-145-5p and passenger strand miR-145-3p act as anti-tumor miRNA in BLCA, thus indicating that both the guide and passenger strands of miRNA have a biological role through the regulation of several genes in BLCA. Their restoration significantly inhibited BLCA cell migration and invasion, therefore possibly downregulating genes that promote metastasis [55].

miR-328-3p is found to be downregulated in BLCA predicting poor prognosis. It acted as an inhibitory miRNA directly targeting ITGA5, via inactivating protein kinase B signaling pathway (PI3K/Akt), consequently inhibiting EMT and tumorigenesis [58]. Another miRNA that inhibits EMT and decreases BLCA invasion through PI3K/Akt signaling pathway is miR-15. B cell-specific Moloney murine leukemia virus integration site 1 (BMI1) is a direct target of miR-15 and via regulating BMI1 through PI3K/Akt pathway, miR-15 inhibited in vivo BLCA cell progression and tumor growth. miR-15 enhanced N-cadherin and Vimentin expression and suppressed E-cadherin levels, thereupon regulating cell invasion and migration by EMT regulation as well as repressed the phosphorylation of AKT expression levels [58]. In T24 and UMUC-3 BLCA, cell lines upregulation of miR-24 inhibited proliferation and induced G1 phase arrest and apoptosis. Its upregulation downregulated the mesenchymal markers including MMP9 amongst others at protein levels, indicating that it can inhibit invasion and EMT. Card-containing MAGUK3 (CARMA3), a tumor suppressor gene required for G protein-coupled receptors (GPCR), shares a link with NF-κB activation. That is of special interest, as downstream targets of NF-κB include cyclin D1, Bcl-2, and MMP9, and thus blocking its activation could inhibit metastasis and induce apoptosis. CARMA3 is negatively regulated and a direct binding target of miR-24 in BLCA with its suppression crucial for miR-24 inhibited proliferation, invasion, and EMT, potentially by downregulating the CARMA3/NF-κB pathway [59].

Compared to normal bladder tissue, miR-23b was found to be downregulated in BLCA and non-malignant cell lines acting as a tumor suppressor. Ectopic expression of miR-23b in J82 and T24 cell lines decreased cell proliferation and colony formation, albeit its re-expression increased cells in the G0/G1 phase while decreasing cells in the S-phase of the cell cycle, triggering G0/G1 arrest and inducing apoptosis. ZEB1, a crucial regulator of EMT and in BLCA responsible for enhanced motility, is a direct target of miR-23b. In BLCA, miR-23b overexpression suppressed the oncogene ZEB1. Thus, miR-23b can mediate EMT, as it post-transcriptionally regulates ZEB1 via targeting its 3′UTR, downregulating its levels [60]. miR-203 in T24 and RT4 cell lines induced cell apoptosis via inhibiting the expression of Bcl-2 and procaspase 3 proteins, while enhancing the expression of Bax and cleaved caspase 3. Twist1 is a direct target of miR-203, with miR-203 acting as a tumor suppressor by negatively targeting Twist1 [61].

Increased ectopic expression of miR-22 increased E-cadherin expression, but decreased that of N-cadherin, vimentin, Snail, Slug, and GSK-3β phosphorylation, therefore suppressing EMT in BLCA acting as a tumor suppressor. MAPK and Snail were found to be direct target genes of miR-22. Overexpression of MAPK1 correlated with poor survival and induced the transcriptional activity of Slug, upregulating vimentin expression, whereas overexpression of miR-22 reversed MAPK or Snail-induced migration and invasion, thus playing an important role in EMT progression and the MAPK1/Slug/vimentin feedback loop. Additionally, knockdown of MAPK1 suppressed Slug and vimentin expression, indicating that Slug possibly acts as a scaffold in MAPK1-induced vimentin expression in BLCA cells. They also performed western blot (WB) assay to reveal the effect of silencing vimentin and found that it suppressed ERK2 phosphorylation, implying that vimentin can activate MAPK1, forming a MAPK1/Slug/vimentin feedback loop in BLCA cells [62].

Another study performed in T24 BLCA cell line examined the expression of TGFβ1/Smad2 following transfection with miR-132 inhibitor. Their results revealed that migration and invasion capabilities, as well as EMT related markers and TGFβ1/Smad2 expression levels, were increased, with a negative expression of Smad2 and miR-132 in BLCA tissue. Additionally, they found miR-132 levels of expression reduced in BLCA tissue with lymph node metastasis. Furthermore, miR-132 suppressed the mesenchymal markers including N-cadherin, Zeb1, Snail, and Vimentin. Hence, miR-132 might suppress EMT via TGFβ1/Smad2 signaling pathway [63]. Through the suppression of HMGA2 expression, miR-484-5p was also found to inhibit metastasis and EMT partly [64]. As mentioned earlier, miRNAs have the unique ability to regulate several protein coding genes, and since BLCA is a heterogeneous disease, when it becomes invasive and metastatic, it substantiates poor prognosis. miR-126 impaired the invasive potential of BLCA by direct downregulation of ADAM9 [66], miR-199a-5p by targeting CCR7 [66], miR-493 by downregulating RhoC and FZD4 [67], and miR-497 by targeting BIRC5 and WNT7A [68]. These miRNAs could present potential therapeutic approaches in treating invasive BLCA.

We formerly stated that the mode of action of miRNAs depends on their target genes in cells or tissues, ergo they not only act as oncosuppressor, but also as oncogenes. One of those oncogenes is miRNA-135a. In colorectal cancer, miR-135a activates the downstream Wnt pathway by suppressing the tumor suppressor gene adenomatous polyposis coli (APC). In BLCA, when compared with adjacent normal tissue, the expression of miR-135a, β-catenin, cyclinD1, vimentin mRNA, and protein expression increased, while GSK-3β and E-cadherin mRNA and protein expression decreased. EMT was induced when miR-135a overexpression inhibited GSK-3β mRNA and protein expression following that activation of Wnt/β-catenin signaling pathway by promoting its related genes including GSK3β, since miR-135a binds specifically to it. Therefore, when miR-135a targets GSK3β, it accelerates BLCA proliferation, migration, and invasion and suppresses apoptosis [37]. Other oncogenic miRNAs are capable of enhancing the expression of EMT transcription factors such as miR-96. TGF-β1 is essential for EMT and cancer progression, and miR-96 induced EMT driven by TGF-β1, which could also regulate the expression of miR-96 target, FOXQ1. miR-96 binds to the 3′untranslated region (UTR) of forkhead box O3 (FOXO3), which triggers apoptosis via regulating genes necessary for cell death, thus inhibiting its function [70]. Furthermore, in RT4 and T24 cell lines, miR-221 and STMN1 were found to be involved in TGF-β1 induced EMT with the expression of miR-221 being upregulated. The microtubule-destabilizing protein, stathmin 1/oncoprotein 18 (STMN1), is an oncogenic protein enhancing invasion and metastasis and during mitosis influences cell cycle progression. STMN1 was downregulated by TGF-β1, and miR-221 suppressed STMN1 expression by targeting 3′UTR. In BLCA, miR-301b levels were overexpressed and induced EMT via early growth response gene 1(EGR1) downregulation [70]. A bioinformatics study found that miR-21 could serve as a prognostic factor for OS and a good indicator of metastasis and tumor recurrence in BLCA with p53, Akt, and PTEN as its target genes [11].

### 2.4. Mechanism of Angiogenesis Inhibition Via miRNA in BLCA

miRNAs have a vital role in angiogenesis by regulating proliferation, differentiation, apoptosis, migration, and tube formation of angiogenesis related cells (Table 4). An imbalance between miRNA regulation and angiogenesis could result in the occurrence and development of cancer and vascular diseases. In BLCA, vascular endothelial growth factor-C (VEGF-C) has an anti-apoptotic and proliferative role, as shown in Figure 3. It is an angiogenic factor that may affect cancer growth, is associated with lymphangiogenesis and regional lymph node metastasis, and can directly act on cancer cell receptors. VEGF is responsible for the growth and permeability of vascular endothelial cells, vasculature, and angiogenesis by inhibiting the apoptosis of endothelial cell lining newly formed vessels. The activation of VEGFR-2 signals through the PI3K/AKT pathway produces the most angiogenic effects accredited to VEGF, as the PI3K/AKT is vital for regulating cellular functions. VEGF-C is a direct target and negatively regulated by miR-128 in BLCA, with it being upregulated while miR-128 expression is downregulated. miR-128 is tissue/cell specific, behaving differently according to its location, but mostly acts as a tumor inhibitor. Its expression suppresses migration and invasion [71]. miR-128 expression in UBC associates with pelvic lymph node metastasis and poor prognosis [72].

The effect of miR-122 on angiogenesis was tested by using chorioallantoic membrane (CAM) system in HT1376 cells. Overexpression of miR-122 decreased HT1376 xenograft tumor growth without significant toxicity and reduced the amount of microvessels in a CAM model. Several targets of miR-122 have been identified including ADAM10 and VEGF-C. miR-122 was found to downregulate VEGF-C expression, inhibiting BLCA growth and angiogenesis via targeting 3′UTR of VEGF-C mRNA. miR-122 inhibited angiogenesis by inhibiting VEGF-C expression, thus blocking Akt/mTOR signaling pathway. Additionally, miR-122 increased BLCA chemo-sensitivity to cisplatin treatment in a VEGFC-dependent manner. Therefore, miR-122 was found to have an inhibitory role in in vivo and in vitro BLCA angiogenesis and growth [12]. AGGF1 is a critical vasculogenesis and angiogenesis factor. In hypoxic conditions, miR-27a mimics significantly decreased the expression level of AGGF1 with up-regulation of miR-27a in UBC. Thus, down-regulation of AGGF1 expression by hypoxia-induced miR-27a expression could possibly represent a pathway for the development of high grade UBC [7]. Additionally, miR-200c has been considered as a regulator of tumor angiogenesis, while AKT2/mTOR is considered a regulator of VEGF and HIF1α. miR-200c was able to negatively regulate the expression of angiogenesis related proteins HIF1α/VEGF expression in BLCA via targeting AKT2/mTOR as did miR-27a, inhibiting angiogenesis and affecting the process of tubule formation [75].

As mentioned earlier, CD44 helps promote invasion and angiogenesis of tumor cells. Following stable transfection of miR-34s precursor, which plays a role in BLCA progression, VEGF and CD44 levels were reduced. In vitro overexpression of miR-34a suppressed angiogenesis by targeting CD44 and reduced tube formation. Thus, miR-34a reduced angiogenesis in BLCA both in vitro and in vivo [52]. CDK4 was a direct target of miR-124, and its overexpression in T24 and 5637 cells increased VEGF expression and impaired the suppressive functions of miR-124 on BLCA angiogenesis, cell viability, and proliferation [76]. miR-153 inhibits in vitro and in vivo T24 and UMUC3 BLCA growth by promoting tumor cell apoptosis, cell migration, invasion, and EMT. In vitro and in vivo assessment of miR-153 expression on BLCA induced HUVEC angiogenesis, and showed that miR-153 suppressed IDO1, a rate limiting enzyme in tryptophan metabolism, which plays a role in tumor cell escape. Ergo, miR-153 suppressed tryptophan metabolism and angiogenesis. Last but not least, STAT3 expression enhances tumor angiogenesis. In BLCA, miR-153 anti-tumor activity was mediated by targeting IDO1, further inactivating IL6/STAT3/VEGF signaling pathway [76].

In NMIBC, urinary miR-214 served as a non-invasive prognostic biomarker of BLCA, distinguishing NMIBC patients from noncancerous controls, and the authors suggested that miR-214 levels could relate to the inhibition of angiogenesis, cell proliferation, and tumor recurrence [73].

## 3. Exosomes and miRNAs in Metastasis and Angiogenesis of BLCA

Exosomes are nanosized membranous vesicles released in subset of extracellular vesicles (EV) at an elevated level in cancer patients and play a vital role in the interaction between tumor cells and TME. In liquid biopsies, miRNA can be detected as free circulating miRNAs or in EV such as exosomes [77]. Exosomes provide a stable source of miRNA, preserving their integrity and protecting them against RNase-mediated degradation. This makes miRNAs stable in liquid biopsy samples, including urine, plasma, and serum, therefore presenting potential biomarkers in non-invasive diagnostic and prognostic methods [78]. Liquid biopsies characterize the genomic landscape of cancer patients and present the most promising diagnostic and prognostic strategy. On the contrary, FDA-approved biomarkers yield a high rate of false positive cases from 12–26%, along with their limited sensitivity when used on their own, thus requiring further testing, including clinical history and microscopy to confirm the result. Additionally, exosomal miRNAs (exomiR) stored at −20 °C remain stable for five years and are resistant to several freeze–thaw cycles, making them well suited for sampling and analysis. Moreover, the increased amounts of circulating exosomes from malignant tissues allow for the identification of several potential exomiR biomarkers, which is an advantage. ExomiRs are able to provide information on several treatment prediction aspects, including their location of origin, specific target, and cellular state. However, exomiRs can affect/induce the development of drug resistance to cytotoxic and molecular target-specific drugs; nevertheless, monitoring and regulating tumor resistance might be possible. Although miRNA panels do present acceptable sensitivities for BLCA detection, they do possess complexities regarding evaluating several markers, hindering their suitability for clinical applications, as they are not cost effective. The difficulties and current limitations in liquid biopsy technologies present a hurdle in terms of isolating and purifying them with limited availability of well-characterized public references for optimizing the collection, storage, and processing of EV-containing body fluids to limit heterogeneity. Additionally, it is not easy to separate tumor derived exosomes, as they originate from both normal and tumor cells and do not have the appropriate genes for normalization of exomiRs expression levels in liquid biopsies established [78,79]. Despite the limitations mentioned, since exosomal research in BLCA is in its early stages, liquid biopsies have been identified as a promising field of cancer research, and it is a matter of time until liquid biopsies replace tissue biopsies in solid tumors.

Contrary to general biomarkers, four of the miRNA panels have sensitivity and specificity above 80% or more, with urine cytology being the most non-invasive test; only three will be covered here, as one study could not be retrieved. The miRNA markers in the panels include miR-99a and miR-125b associating with tumor grade, with a sensitivity of 86.7%, a specificity of 81.1%, and a positive predicted value (PPV) of 91.8% [80]. The second panel used miRNA-96 as molecular marker with a sensitivity of 72.3%, and a specificity of 88.9%, but the sensitivity of urine cytology when combined with miRNA-96 improved to 79.8% [81]. The third study identified a 25-target diagnostic signature that predicted the presence of BLCA with a sensitivity of 87% and a specificity of 100% [82]. A micro-fluidic chip-based system was used to analyze EV from urine samples of normal and BLCA patients and resulted in 81% sensitivity and 90% accuracy in diagnosing BLCA [6]. Therefore, we will not just be talking about FDA-approved biomarkers, but rather the role of exosomes in BLCA in relation to miRNAs, since in cancer, miRNA secretion pathways are dysregulated, making miRNA an attractive non-invasive candidate molecules for liquid biopsies.

In tumor progression of metastatic cancer cells, exosomal secretion of miRNAs increases [83]. In a study by Baumgart et al., they analyzed the miRNA expression in exosomes secreted by BLCA from different degrees of invasiveness. Fifteen miRNAs were significantly altered in exosomes of invasive BLCA compared to their non-invasive counterparts and characterized by a specific miRNA expression pattern. Seven miRNA were upregulated and eight downreglulated in exosomes of invasive cells as well as the invasive cells when compared to their non-invasive counterpart. The expression of the following miRNAs was significant (*p* < 0.05): miR-30a-3p, miR-99a-5p, and miR-137-3p were up-regulated, and two were down-regulated, miR-141-3p and miR-205-5p. In vivo analysis from patient samples revealed the cellular expression of miR-141-3p, miR-200a-3p, and miR-205-5p was significantly down-regulated in MIBC tumors (*p* < 0.05) compared to NMIBC. Therefore, miRNAs are differentially expressed in invasive cells as well as their exosomes in comparison to their non-invasive counterparts; thus, the molecular content of the exosomes is at least partially similar to that of the host cell, reflecting their cellular property. miR-200 family, as previously discussed, associates with EMT in BLCA, confirming the theory of increased invasiveness following loss of adhesion. Some miRNAs are selectively packaged into exosomes such as miR-30a-3p. miR-30a-3p was deregulated only in exosomes of invasive BLCA cells, not in their parental nor non-invasive counterparts, which possibly plays a role yet to be researched to see if this particular modification influences the TME [84].

RAB27A and RAB27B were found to be regulators of exosomal miRNA secretion in BLCA, with overexpression of RAB27B correlating with poor prognosis of BLCA. In breast cancer ER-positive patients, higher RAB27B expression correlates with lymph node metastasis. miR23b has a tumor suppressor function, and its intracellular level and function were influenced by exocytosis. This influence was selective and acted as a mechanism to coordinate activation of a metastatic cascade [85]. Another microRNA molecular profiling study examined the co-occurrence of miRNA profiles in BLCA patients from four matched bio-specimen sources: tumor tissue, plasma, urine exosomes, and white blood cells (WBC). They found that a significant number of miRNAs are identifiable in urine exosomes and WBCs of the same patients, however not in blood plasma [86].

These results present an understanding of the physiological functions of exosomes in BLCA and how they modulate aspects in cancer biology. Since miRNAs travel via exosomes in the TME and change the phenotype of neighboring cells, they could be used to develop non-invasive tools of BLCA, as BLCA cells and urine present high chances for exosome secretion directly into the fluid.

## 4. Conclusions

The studies summarized in this review clearly indicate that in bladder cancer, microRNAs are involved in the regulation of metastasis and angiogenesis. We highlighted the role of several miRNAs in controlling the activity of major cancer-related signaling molecules. The emerging picture of metastasis and angiogenesis in tumor etiopathogenesis is one of increasing complexity, as versatile regulator miRNAs play pivotal roles in each hallmark of tumor progression. Strategies targeting molecules that enable cancer cells to metastasize or employ anti-angiogenic therapy along with immune modulation could possibly alter the balance of the tumor microenvironment, allowing tissue repair and relieving symptoms of autoimmunity. With the rapid development of sequencing technologies allowing for the discovery of dysregulated miRNAs, identifying the role of miRNAs in BLCA tumor could possibly promote the development of new markers as diagnostic and prognostic tools.

Nonetheless, there is strong evidence that miRNAs control the activity of major cancer related signaling molecules. There has been a rapid development of miRNA antagonists, mimics, and delivery technologies, allowing the use of miRNAs in metastatic tumor therapy of other cancers. Therefore, identifying aberrant miRNA expression and the oncogenic or tumor-suppressive targets is inevitable for the clinical development of novel cancer therapeutics.

## Figures and Tables

**Figure 1 cancers-13-00891-f001:**
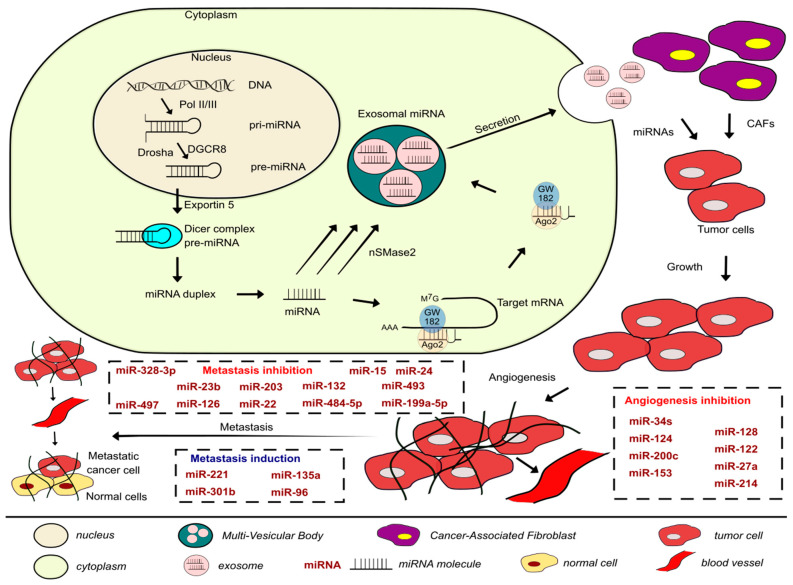
Development of the metastatic cascade in BLCAs along with the miRNAs involved in inhibiting or promoting the process covered in this paper and a brief description of the biogenesis, packaging, and secretion of exosomal miRNAs.

**Figure 2 cancers-13-00891-f002:**
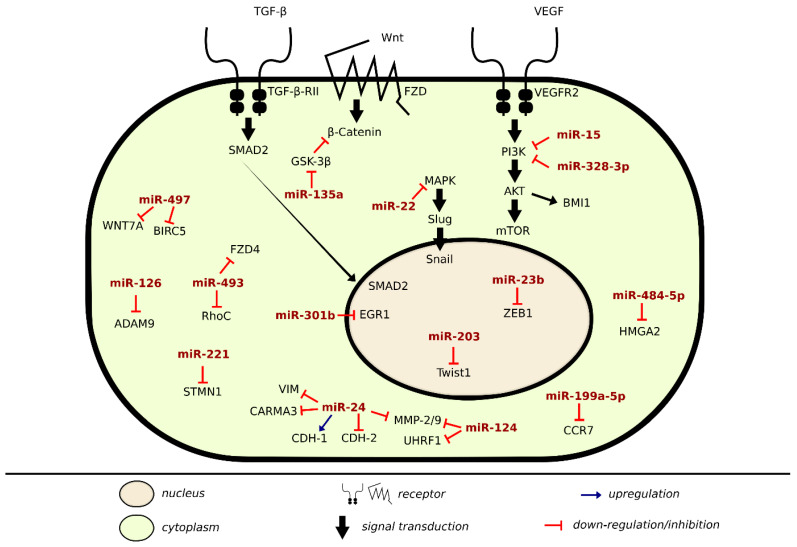
Metastatic inhibition and induction by miRNA in BLCA.

**Figure 3 cancers-13-00891-f003:**
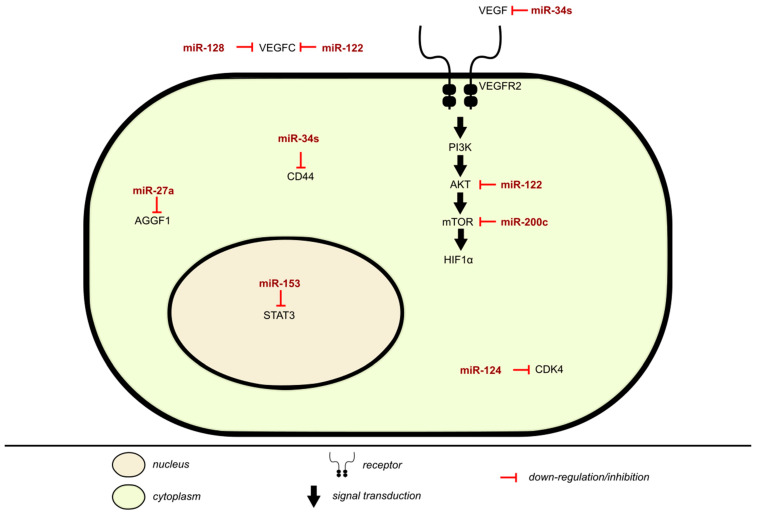
Inhibition of angiogenesis by miRNA in BLCA.

**Table 1 cancers-13-00891-t001:** Functions of miRNAs with CAMs in BLCA.

miRNA	Signal Transduction Molecules	Target Cytoskeletal Protein	Cell–Cell Adhesion Molecules	Cell–Matrix Adhesion Molecules	Function	Notes	References
miR-373	It inhibits expression of E-cadherin downstream genes: cyclinD1, c-myc, MMP2 mRNA levels		Its expression positively associates with		Inhibits cell proliferation and metastasis by activating E-cadherin expression. Acted as a tumor suppressor	Expression significantly correlated with tumor stage, grade, and lymph node metastasis	[34]
			E-cadherin, activating its expression by interacting with E-cadherin promoter				
miR-141 and miR-200b	Activity of MMP-2, MMP-9 enzymes was inversely proportional to that of miRNAs	Vimentin levels were downregulated following miRNA overexpression	E-cadherins levels increased, while		Loss of expression increased invasion and migration capacity by up-regulating N-cadherin and downregulating	Urine analysis distinguished presence of or lack thereof patient lymph node metastasis	[35]
			N-cadherin levels decreased when both miRNAs were overexpressed		E-cadherin		
miR-145	MMP-9		N-cadherin		Suppresses migration and invasion by targeting and regulating N-cadherin and MMP-9 expression		[36]
miR-135a	Increased expression of β-catenin, cyclinD1, downregulation of GSK3β	Increases expression of vimentin	Decreases expression of E-cadherins		It is an oncogene that accelerates EMT, migration, and invasion via activation of Wnt/β-catenin signaling pathway and inhibiting GSK3β		[37]
miR-200b and miR-200c	Decreases ZEB1, ZEB2, EGFR, ERRFI-1 expression		Increases E-cadherin expression		Control the EMT process, decreasing cell migration, and via E-cadherin affects response to EGFR therapy		[38]
miR-3619	It interacts with p21 promoter, targeting β-catenin and CDK2. Downregulates Snail protein levels	Decreases vimentin expression	Increases E-cadherins and decreased N-cadherin expression		Inhibits Wnt-β-catenin signal pathway and EMT progression	Low expression of p21 and miR-3619 linked to poor OS.	[39]
						β-catenin and CDK2 are direct downstream targets of miR-3619	
miR-31				Integrin α5 is a direct target of miR-31	miR-31 acted as a tumor suppressor deactivating Akt and ERK. Activation of miR-31/ITGA5 axis increased sensitivity of UBC to mitomycin-C	Downregulation of miR-31 correlates with higher risk of recurrence and progression of noninvasive UBC cases	[40]
miR-124-3p	FAK/PI3K/AKT and FAK/Src signalling pathways			Integrin α3 is a direct target of miR-124-3p	By targeting ITGA3 and downstream FAK/PI3K/AKT and FAK/Src signaling pathways, miR-124-3p suppresses cell migration and invasion		[41]
miR-34a	CD44 a target gene, inhibits β-catenin expression	Inhibited vimentin expression	Promotes E-cadherins and inhibited N-cadherin expression		Anti-metastatic and suppresses angiogenesis by directly targeting CD44		[42]

**Table 2 cancers-13-00891-t002:** Major ECM molecules regulated by miRNAs in BLCA.

miRNA	Target	Function	References
miR-142-3p	ADAM9 (direct)	ADAM9 expression was regulated in wing and leg mesenchymal cells and contributed to the modulation of position-dependent chondrogenesis.	[44]
miR-200b	MMP-16 (direct)	miR-200b overexpression inhibited TGF-β1-induced MMP-16 upregulation and BLCA migration	[54]
miR-370-3p	Wnt7a	Wnt7a overexpression up-regulated MMP1/10 to degrade the extracellular matrix and to facilitate UBC cell invasion	[42]
miR-139-5p and miR-139-3p	MMP11	MMP11 contributed to migration and invasion through its regulation of many oncogenic genes. Several genes known to contribute to cancer cell aggressiveness were downstream from MMP11, such as CXCL1 and CXCL3	[55]

**Table 3 cancers-13-00891-t003:** miRNAs involved in metastasis induction/inhibition in BLCA.

microRNA	Samples/Cell Culture	Targets/ Regulators	Function	Patient’s Prognosis	References
miR-124	Human bladder transitional cancer cell lines J82 and T24	UHRF1	miR-124 can impair the proliferation or metastasis of human BLCA cells by down-regulation of UHRF1		[56]
miRNA-139-5p and miRNA-139-3p	human BLCA cell lines: T24 and BOY	MMP11	Downregulation of both miRNAs enhanced BLCA cell migration and invasion	Higher expression of MMP11 predicted shorter survival of BLCA patients (*p* = 0.029)	[55]
pre-miR-145 and miR-145-5p	Clinical tissue: 62 BLCA patients at Kagoshima University Hospital between 2003 and 2013	UHRF1	Overexpression of miR-124 in vitro, attenuated cellular proliferation, migration, invasion, and angiogenesis by downregulating UHRF1		[55]
miR-328-3p	Cell lines: Urinary epithelial SV-HUC-1 and 5637, T24, J82 BC cell lines Clinical tissue:28 pairs of BLCA tissues and adjacent normal samples were acquired from the Yinzhou Hospital	ITGA5	miR-328-3p inhibited the development of BLCA by targeting ITGA5 and inactivating the PI3K/AKT pathway	Downregulation of miRNA predicted poor prognosis in BLCA patients	[57]
miR-15	T24, BIU87, HT1376 BLCA cell lines, and normal uroepithelial cell lines SV-HUV-1	BMI1	Overexpression of miR-15 inhibited EMT and PI3K/AKT pathway		[58]
miR-24	Human BLCA cell lines T24, UMUC-3, J82, 5637 and normal transitional epithelial cell line SV-HUC-1	CARMA3	Upregulation of miR-24 inhibited proliferation, invasion, EMT, and induced apoptosis of T24 and UMUC-3 cells. Additionally, upregulation of miR-24 decreased the protein levels of cyclin D1, CDK4, CDK6, p-Rb, and Bcl-2		[59]
miR-23b	T24 and J82, and normal SV-HUC-1	3′UTR of Zeb1	Overexpression of miR-23b suppressed the oncogene ZEB1 suppressing cell proliferation, invasion apoptosis, and cell cycle arrest	Patients with higher miR-23b expression had longer OS compared to patients with low miR-23b expression. Additionally, its expression distinguished malignant from normal tissues	[60]
miR-203	T24, RT4, normal urothelial cell line SV-HUC-1	Twist1	miR-203 mimic significantly reduced BLCA cell proliferation, migration, and invasion, and induced apoptosis by targeting Twist1		[61]
miR-22	T24, UM-UC-3, as well as one normal bladder cell line SV-HUC-1	MAPK1 and Snail	miR-22 was found to suppress cell proliferation/apoptosis by directly targeting MAPK1 and inhibiting cell motility by targeting both MAPK1 and Snail	Low-expression of MAPK1 or Snail is an independent prognostic factor for better OS	[62]
miR-132	Human BLCA cell lines T24 and human normal urothelial cell line SV-HUC-1 Clinical tissue:32 patient samples	SMAD2	miR-132 may play a suppressive role in the metastasis of BLCA cells via TGFβ1/Smad2 signaling pathway. Overexpression of miR-132 suppressed expression of mesenchymal cell markers (N-Cadherin, Zeb1, Snail, and Vimentin)		[63]
miR-484-5p	Cell lines: Human bladder cancer cell lines SW780, T24, HT1376, and HT5637 and human bladder epithelial cell lines HU609 and HEK293 cells Clinical tissue:15 patient samples who had undergone surgery and primary therapy	HMGA2	miR-485-5p exerts a suppressive effect, partly through the suppression of HMGA2		[64]
miR-126	Cell lines: HUC, EJ138, MCF10A, TCCSUP, J82, and 293FT cells Clinical samples: TaLG (*n* = 3), TaHG (*n* = 3), CIS (*n* = 3), T1LG (*n* = 3), T1HG (*n* = 3), and T2HG (*n* = 6)	ADAM9	miR-126 exerts its tumor suppressive role by targeting ADAM9 to inhibit cell invasion		[65]
miR-199a-5p	T24 and SV-HUC-1 cell lines Clinical samples:40 BLCA tissue samples and adjacent normal tissue from patients who underwent transurethral bladder tumor resection or radical cystectomy	3′UTR of CCR7	miR-199a-5p was confirmed to be able to target the 3′ UTR of CCR7 and regulate the expression of CCR7, MMP-9, and vimentin and E-cadherin		[66]
miR-493	SV-HUC-1, T24, J82, and TCCSUP cells Clinical samples: human bladder cancer tissue array from US Biomax	RhoC and FZD4	miR-493 possibly a tumor suppressive, inhibiting cell invasion and migration by blocking FZD4 and RhoC, implicating the Wnt-PCP pathway in bladder carcinogenesis		[67]
miR-497	BOY and T24 cell lines Clinical samples: 5 BLCA patients and five normal epithelial samples of patients who underwent cystectomy or transurethral resection.	BIRC5 and WNT7A	Downregulation of miR-195/497 contributed to BLCA progression and metastasis		[68]
Induction					
miRNA-135a	Normal human SVHUC-1 epithelial cells, EJ, T24, BIU87, ScaBER, and 5637165 Clinical samples: paired BLCA tissues and adjacent normal tissues were obtained from patients with BLCA who had undergone a bladder resection	GSK3β	miR-135a accelerates the EMT, invasion, and migration of BLCA cells by activating the Wnt/β-catenin signaling pathway through the downregulation of GSK3β expression.		[37]
miR-96	HT1376	FOXQ1	TGF-β1 could change the expression of FOXQ1 induced by miR-96, which revealed that TGF-β1 regulates miR-96/FOXQ1 signaling		[69]
miR-221	RT4 and T24	3′UTR of STMN1	miR-221 can facilitate the TGFβ1-induced EMT process in human BLCA cells by suppressing STMN1		[70]
miR-301b	J82, UM-UC-3, T24, 5637 BLCA cell line and normal SVHUC-1 cell line Clinical tissue: 31 paired BLCA and normal tissue obtained from patients who underwent radical cystectomy.	EGR1	miR-301b promotes the proliferation, migration, and aggressiveness of human BLCA cells by inhibiting the expression of EGR1.		[70]

**Table 4 cancers-13-00891-t004:** miRNAs involved in angiogenesis inhibition in BLCA.

microRNA	Samples/Cell Culture	Targets/Regulators	Function	References
miR-128	Human bladder epithelial cell line SV-HUC-1, and the BLCA cell lines T24, 5637, 3-UM-UC-3, and RT4	VEGF-C	Overexpression of miR-128 inhibited growth rate, proliferation, migration, and invasion capacities.	[71,72]
miR-122	BLCA cells BIU-87, T24, SW780, HT1376, 5637, RT4, and normal bladder epithelial cell line SV-HUC-1	3′-UTR of VEGF-C	miR-122 regulated cell proliferation through the VEGFC/AKT/mTOR signaling pathway.	[12]
miR-27a	Clinical samples: Urothelial carcinoma and normal urothelial tissue samples were collected from 59 patients undergoing surgery	AGGF1	Down-regulation of AGGF1 expression by hypoxia-induced miR-27a expression represents a signaling network for development of high-grade UBC.	[7]
miR-214	138 patients with primary urothelial carcinoma of the urinary bladder and 144 healthy controls		The urinary levels of cell-free miR-214 were significantly higher in the NMIBC patients than in the controls. Thus, they could be used as a prognostic marker for NMIBC.	[73]
miR-34s	Cell lines: 5637, T24, HT-1376, J82, SCABER, and EJ Clinical samples: BLCA tissue and adjacent normal tissue specimens	CD44	miR-34a overexpression can inhibit bladder cell migration, invasion, tube formation in vitro, and metastasis and angiogenesis in vivo. Additionally, CD44-mediated functions can be reversed by miR-34a in bladder cells.	[52]
miR-124	Hek293, human normal cell SV-HUC-1 and BLCA T24, 5637, J82, and UM-UC-3 Clinical tissues: 83 bladder tissues and their adjacent non-tumor tissues	CDK4	Overexpression of miR-124 induced by mimic transfection was observed to inhibit the cells viability, angiogenesis, and proliferation.	[74]
miR-200c	Human bladder epithelial cell line, SV-HUC-1, BLCA cell lines 5637 and T24	Akt2	miR-200c could suppress HIF-1α/VEGF expression in BLCA cells and inhibit angiogenesis, and these regulations were achieved by targeting Akt2/mTOR.	[75]
miR-153	T24, UMUC3, 5637, and J82 cell lines and an immortalized human normal bladder epithelial cell line SV-HUC-1 Clinical tissue: normal and cancerous tissue specimens from 45 BLCA patients	IDO1	IDO1 mediated miR-153 anti-tumor activity in BLCA via inactivating the IL6/STAT3/VEGF pathway.	[76]

## Data Availability

No new data were created or analyzed in this study. Data sharing is not applicable to this article.

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
