# Peer review of "MicroRNAs: Their Role in Metastasis, Angiogenesis, and the Potential for Biomarker Utility in Bladder Carcinomas"

_cancers, 2021, doi:10.3390/cancers13040891_

Round 1

Reviewer 1 Report

This study summarize the roles of 32 various miRNAs in BLCA angiogenesis and metastasis. The study is well-organized and well-written. I only have several questions as followed.

  1. In line 79, is there any reference to support this sentence?
  2. The format of in vivo and in vitro should be Italic.
  3. What is the meaning of black line on tumor cells in figure 1?

Author Response

We appreciate the time and effort that you and the reviewers have dedicated to providing your valuable feedback on our manuscript. And we thank you for giving us the opportunity to submit a revised draft of our manuscript titled: ”MicroRNAs: Their Role in Metastasis, Angiogenesis, and The Potential for Biomarker Utility in Bladder Carcinomas”.

We are grateful to the reviewers for their insightful comments on our paper and have been able to incorporate changes to reflect most of the suggestions provided by the reviewers. We have highlighted the changes within the manuscript.

Below is a point-by-point response to your comments.

  • Comment 1: In line 79, is there any reference to support this sentence?

Response: Thank you for mentioning it, there was a typo with the one of author names Matz instead of Motz. We fixed it now (page 2 line 88) and this is the reference: “Motz GT, Coukos G. The parallel lives of angiogenesis and immunosuppression: cancer and other tales. Nat Rev Immunol. 2011 Oct;11(10):702–11”

  • Comment 2: The format of in vivoand in vitro should be Italic.

Response: Thank you for bringing it up, we have fixed them and they are tracked throughout the paper.

  • Comment 3: What is the meaning of black line on tumour cells in figure 1?

Response: The arrows are shown following angiogenesis, thus representing the formation of a tumour-promoting, immunosuppressive, metastatic environment facilitating tumourigenesis and tumour development i.e. remodelling of tumour microenvironment.

Reviewer 2 Report

This is a well written reviewarticle about the known function of MicroRNAs in Metastasis and Angiogenesis. It is informative buit contains a initial wrong assumption as you give the statement that BLCA is an aggressive epithelial tumorwith a high rate of early systemic dissemination. For a clinical working urologist this is wrong as 80-85% of bladder carcinomas are superficial and these tumors are really not very dangerous but they are malignant. For me it is not well understood why most of the papers dealing with MicroRNAs do not discriminate between superficial and invasive bladder tumors. From the biological behavior they are different and they must be differnt too in the expression of factors which facilitate metastasis and angopgenesis. The work up of diffferences between these two groups should give more inside knowledge in teh development of aggressive tumors. Your work up in groups of cadherins and molecular regulation is interesting and informative.

Author Response

We appreciate the time and effort that you and the reviewers have dedicated to providing your valuable feedback on our manuscript. And we thank you for giving us the opportunity to submit a revised draft of our manuscript titled: ”MicroRNAs: Their Role in Metastasis, Angiogenesis, and The Potential for Biomarker Utility in Bladder Carcinomas”.

   We are grateful to the reviewers for their insightful comments on our paper and have been able to incorporate changes to reflect most of the suggestions provided by the reviewers. We have highlighted the changes within the manuscript.

I was very interested to hear your feedback as a clinical urologist. I have made a note to discuss your observation on the difference between superficial and invasive tumours with my colleagues so that we can take this into account in future research.

We have reworded the statement on page 2 (lines 67-69) to reflect that here in particular we are referring to the muscle invasive BLCA as the aggressive type, as you and Dr Cora Sternberg (the paper we referenced doi: 10.1093/annonc/mdl231) stated that one third of the patients with MIBC develop a locally invasive or metastatic disease which can infiltrate the musculature, which depending on the pathologic stage and grade has a low (25-35%) 5 year survival rate.

We have also added a section in the introduction (page 2, lines 57-75) to address the pathophysiology and molecular biology of the categories of bladder cancer: NMIBC and MIBC. Furthermore, we tried were possible in the text to refer the different types of BLCA; NMIBC or MIBC (pages 3, lines 119; page 16 line 737). As you mentioned there isn’t much in the literature addressing miRNAs in the superficial bladder tumours, and we will apply your recommendation in our future work. We made this assumption in discussing the roles of the miRNAs here because we are discussing metastatic thus more aggressive BLCA but as you mentioned the superficial non-invasive type is not very dangerous, with distant metastasis a very rare occurrence and can be treated with local therapy or transurethral resection depending on the tumour stage.

Reviewer 3 Report

This review very thoroughly assesses the state of the field of angiogenesis and metastasis in bladder cancer, with an emphasis on microRNAs. They discuss microRNAs as diagnostic biomarkers, as well as the potential for targeting these molecules therapeutically. The inclusion of exosomes is valuable. Overall the work is an impressive compilation of the literature and should be a very helpful resource to the bladder cancer research community.

Introduction – Please mention the high proportion of non-muscle invasive disease vs. muscle invasive in the population and, thus the potential for early diagnostic biomarkers (e.g. miRNAs) to catch the disease before metastasis. This makes bladder cancer a uniquely good opportunity for biomarker development.

Grammar:  p.3 L138 – sentence fragment; p.6 L200 grammar; p.7 L240 period;

Figures 1,2,3 nicely depict and summarize the role of various miRNAs in angiogenesis and metastasis, which are discussed in great detail in the text.

p.10 L421 missing a reference?

p.16 L697 what fluid are the exosomes from e.g. blood or urine?

Table 1 – decrease line spacing and add vertical column lines for readability.

Author Response

We appreciate the time and effort that you and the reviewers have dedicated to providing your valuable feedback on our manuscript. And we thank you for giving us the opportunity to submit a revised draft of our manuscript titled: ”MicroRNAs: Their Role in Metastasis, Angiogenesis, and The Potential for Biomarker Utility in Bladder Carcinomas”.

We are grateful for the insightful comments on our paper and have been able to incorporate changes to reflect most of the suggestions provided by the reviewers. We have highlighted the changes within the manuscript.

Below is a point-by-point response to your comments.

  • Comment 1: Introduction – Please mention the high proportion of non-muscle invasive disease vs. muscle invasive in the population and, thus the potential for early diagnostic biomarkers (e.g. miRNAs) to catch the disease before metastasis. This makes bladder cancer a uniquely good opportunity for biomarker development.

Response: Thank you for pointing this one out indeed it is essential to be mentioned. We have modified the introduction and included it in (p.2 L57-75) and added a few sentences in (p.3, L119-123).

  • Comment 2: Grammar:  p.3 L138 – sentence fragment; p.6 L200 grammar; p.7 L240 period;

Response: -sentence fragment has been fixed (p.4 L155): The functional loss of E-cadherin gene promoter is a hallmark of EMT, which is detected in distant metastases, and in humans the E-cadherin promoter; the E-box elements are responsible for its transcriptional repression in non-E-cadherin-expressing mesenchymal cells.

- p.6 L200 grammar: we changed it (p.6 L253): “Hynes et al found that activation of TGFβ alone without the presence of platelets was unable to produce effects with a magnitude similar to that of platelets (28)”

-p.7 L240 period: we added a full stop after “gene expression” (p.6 L294), “Loss of E-cadherin reprograms global gene expression. This promotes relocalisation of β-catenin which a transcriptional co-activator from the nucleus to the plasma membrane (34).”

  • Comment 3: Figures 1,2,3 nicely depict and summarize the role of various miRNAs in angiogenesis and metastasis, which are discussed in great detail in the text.

Response: Thank you very much, it really is encouraging to hear this.

  • Comment 4: p.10 L421 missing a reference?

Response: I seem to have an issue with the formatting and the lines might be slightly different, so just double checking this is the sentence you are referring to ” where in immunocompetent mouse models blockade of IL-6 decreased CD44 expression attenuating tumour aggressiveness (53).”  

If so then this is the reference is (53) Wu C-T, Lin W-Y, Chen W-C, Chen M-F. Predictive Value of CD44 in Muscle-Invasive Bladder Cancer and Its Relationship with IL-6 Signaling. Ann Surg Oncol. 2018 Nov;25(12):3518–26. doi: 10.1245/s10434-018-6706-0

  • Comment 5: p.16 L697 what fluid are the exosomes from e.g. blood or urine?

Response: In this particular case we were referring to urine:urinary exosomes.

  • Comment 6: Table 1 – decrease line spacing and add vertical column lines for readability.

Response: We agree and added the vertical lines but originally we wanted it to be presented in a separate file with the orientation set as landscape but we have modified it now.

Reviewer 4 Report

Overall the article was very well written and organized. It provides a comprehensive account of miRNAs and their regulatory role in metastasis and angiogenesis during the development of bladder cancer (BLCA). It also describes the potential usage of miRNAs as biomarkers for BLCA.

The authors have nicely illustrated the roles of various miRNAs involved in different stages, cellular processes/pathways associated with BLCA. It provides a thorough summary of findings reported in the literature around miRNAs with a focus on BLCA.

In view of the above, the article will serve as a guide to enhance understanding and applications of miRNAs in BLCA and other cancers.

Minor comments:

  • Given the increasing interest in the utility of liquid biopsy-based methods, it would be good if authors can add a paragraph describing the pros and cons of applications of miRNAs as biomarkers in liquid biopsy-based detection/diagnosis/prognosis of BLCA.
  • line no. 27-28: It seems that beginning of the sentence missing some connecting words. "However, care needs be as the ..." 
  • line no. 144: Chaffer et al instead of Weinberg et al (else please update the reference accordingly)

Reference no. 20:

Chaffer CL, San Juan BP, Lim E, Weinberg RA. EMT, cell plasticity and metastasis. Cancer Metastasis Rev. 2016 808
Dec;35(4):645–54.

Author Response

We appreciate the time and effort that you and the reviewers have dedicated to providing your valuable feedback on our manuscript. And we thank you for giving us the opportunity to submit a revised draft of our manuscript titled: ”MicroRNAs: Their Role in Metastasis, Angiogenesis, and The Potential for Biomarker Utility in Bladder Carcinomas”.

We are grateful for the insightful comments on our paper and have been able to incorporate changes to reflect most of the suggestions provided by the reviewers. We have highlighted the changes within the manuscript.

Below is a point-by-point response to your comments.

  • Comment 1: Given the increasing interest in the utility of liquid biopsy-based methods, it would be good if authors can add a paragraph describing the pros and cons of applications of miRNAs as biomarkers in liquid biopsy-based detection/diagnosis/prognosis of BLCA.

Response: This is really valuable as it is a growing field so thank you for pointing it out we have modified (page16 and 17, lines 741-787)

  • Comment 2: line no. 27-28: It seems that beginning of the sentence missing some connecting words. "However, care needs be as the ..." 

Response: Thank you for pointing it out, we have reworded (page 1, line 32) it and hope it sounds better now. We meant to refer to the previous sentence in specific “ altering the expression of miRNA” thus care needs to be taken in terms of altering the expression.

“Precaution must be taken as the complexity of miRNA regulation might result in targeting several downstream tumour suppressors or oncogenes enhancing the effect further.”

  • Comment 3: line no. 144: Chaffer et al instead of Weinberg et al (else please update the reference accordingly)

Reference no. 20:

Chaffer CL, San Juan BP, Lim E, Weinberg RA. EMT, cell plasticity and metastasis. Cancer Metastasis Rev. 2016 808
Dec;35(4):645–54.

Response: We are terribly sorry and apologise for the mistake, we corrected it.